# A Legionnaires’ Disease Cluster in a Private Building in Italy

**DOI:** 10.3390/ijerph18136922

**Published:** 2021-06-28

**Authors:** Maria Luisa Ricci, Maria Cristina Rota, Maria Grazia Caporali, Antonietta Girolamo, Maria Scaturro

**Affiliations:** Department of Infectious Diseases, Istituto Superiore di Sanità, 00161 Rome, Italy; marialuisa.ricci@iss.it (M.L.R.); mariacristina.rota@iss.it (M.C.R.); mariagrazia.caporali@iss.it (M.G.C.); antonietta.girolamo@iss.it (A.G.)

**Keywords:** *Legionella*, typing, private homes

## Abstract

Legionnaires’ disease (LD) is a severe pneumonia caused by bacteria belonging to the genus *Legionella*. This is a major public health concern and infections are steadily increasing worldwide. Several sources of infection have been identified, but they have not always been linked to human isolates by molecular match. The well-known *Legionella* contamination of private homes has rarely been associated with the acquisition of the disease, although some patients never left their homes during the incubation period. This study demonstrated by genomic matching between clinical and environmental *Legionella* isolates that the source of an LD cluster was a private building. Monoclonal antibodies and sequence-based typing were used to type the isolates, and the results clearly demonstrated the molecular relationship between the strains highlighting the risk of contracting LD at home. To contain this risk, the new European directive on the quality of water intended for human consumption has introduced for the first time *Legionella* as a microbiological parameter to be investigated in domestic water systems. This should lead to a greater attention to prevention and control measures for domestic *Legionella* contamination and, consequently, to a possible reduction in community acquired LD cases.

## 1. Introduction

*Legionella pneumophila* is an opportunistic premise plumbing pathogen, able to persist and grow in plumbing of any kind of building. Although the disease is caused by about a half of the 62 known species belonging to the Gram-negative bacteria of the genus *Legionella*, the species pneumophila, and mainly serogroup 1, is the most commonly found in cases of disease. When drops of contaminated aerosols produced by any man-made devices, such as showers or decorative fountains or cooling towers, are inhaled *Legionella* can cause either a severe form of pulmonary infection, named Legionnaires’ disease (LD) or a milder one, named Pontiac fever. Case fatality rate in Europe in 2018 was 8%, but higher percentages between 15 and 34% are reported for more susceptible patients and among hospitalized cases [1,2]. Indeed, immunocompromised individuals and elderly are at higher risk of infection and males more than females, with a ratio 2.8:1. As largely documented and reported by LD surveillance schemes in place in many countries around the world, most reported LD cases are often part of epidemic clusters and are more often due to the contamination of the water systems of hospitals or hotels [1]. 

Epidemiological and microbiological investigations are more likely to successfully allow identification of the source of infection when cases are nosocomial acquired and travel associated. Community cases represent the highest percentage of *Legionella* infections, mostly documented as outbreaks due to contamination of cooling towers [3,4]. However, it has been largely reported that a huge number of sporadic community cases remain without an ascertained source of infection. In Italy, among 3199 cases notified in 2019, 84.1% were classified as community acquired not having been defined the origin of infection and because the patients did not stay away from home in the 2–10 days preceding the onset of symptoms [5].

Although many environmental investigations have demonstrated the undetectability of *Legionella* in municipal water networks, the contamination of private homes is well known and some domestic cases, especially in immunocompromised patients, have been also documented [6]. Dufresne et al. (2012) showed that out of 36 sporadic cases of community acquired LD, 33% had a *Legionella*-contaminated home, although only 14% of environmental and human *Legionella* isolates had a similar genomic profile, as shown by restriction fragment length polymorphism (RFLP) and pulsed-field gel electrophoresis (PFGE) [7]. In Italy, a multicentric study demonstrated that about 22% of the hot domestic water samples were *Legionella* spp. positive, and *L. pneumophila* was identified in 75% of them [8].

Here, we reported the occurrence of a cluster of two LD community acquired cases in people living in two different apartments of the same building, where the household plumbing was demonstrated to be contaminated by *Legionella pneumophila*.

## 2. Materials and Methods

### 2.1. Cases

An 81-year-old man and a 73-year-old woman, both residents in a municipality of Northern Italy, were hospitalized on 28 and 29 May 2017, respectively, with pneumonia symptoms. The patients lived in apartments located, respectively, on the first and second floors of the same building, and, as reported in their notification form, both had stayed only in their own home.

The man was affected by arterial hypertension and ischemic heart disease and unfortunately died two days after admission, while the woman was not affected by underlying diseases.

### 2.2. Diagnosis and Culture

Diagnosis of *Legionella* pneumonia was made by radiological exam and urinary antigen test in both patients. Additionally, in order to obtain the causative pathogen, cultural examination of lung tissue fragments was required after autopsy. Lung tissue fragments were homogenized and aliquots of them were plated on selective and non-selective BCYE agar plates and incubated at 37 °C for a maximum of 10 days. Suspected colonies were subcultured on BCYE agar plates with and without cysteine and those grown on medium with cysteine were tested by latex agglutination test (Oxoid) for definitive confirmation of serogroup 1.

### 2.3. Environmental Investigation

Soon after hospital admission, an environmental investigation and water sampling were carried out. Overall, nine water samples were collected as shown in Table 1: two samples were collected in each of the patients’ apartments, one in the apartment of a third person with a suspicious pneumonia, one from the boiler and one from the recirculating water of the building, and two from the municipal water network. All water samples were analyzed by culture according to the ISO 11731:2017, using filtration with washing procedure and considering the samples as water with high background.

### 2.4. Typing

*Legionella pneumophila* serogroup 1 colonies were typed to establish monoclonal subgroup by MAb Dresden panel and sequence type (ST) by Sequence Based Typing (SBT) [9,10].

## 3. Results

The two LD cases were clinically and laboratory confirmed. Urinary antigen test was positive in both patients and cultural isolation of *Legionella pneumophila* serogroup 1 was obtained by lung tissue fragments of the patient who died. Several colonies were typed subgroup Philadelphia ST23.

Six water samples collected in the patients’ apartments and from the boiler and recirculation hot water were positive for *Legionella pneumophila* serogroup 1 with a contamination ranging from 1 × 10^3^ to 1.4 × 10^4^. The samples collected from the municipal water network and the apartment of a third suspected case were negative (Table 1).

*Legionella* colonies isolated from environmental samples were serogroup 1 by latex agglutination test and all were typed subgroup Philadelphia ST23 as the clinical samples, demonstrating the epidemiological relationship between clinical and environmental *Legionella* strains.

## 4. Discussion

This study describes a cluster of two cases of LD in which the source of the infection is, for both, the water system of their apartments, located in the same building. Colonization of home showers has often been reported and, although the risk of infection is generally higher among the immunocompromised and the elderly, home showers should be checked and monitored more frequently [11].

The two LD cases here described, and a third suspected case, in the absence of a match between clinical and environmental strains, are usually classified as community acquired cases, with no known source of infection.

In this case, after having identified the source of the infection, the Local Health Authorities recommended the disinfection of the system, the raising of the hot water temperature and provided indications for a correct management of the water system.

In Germany, since 2011, due to the high incidence of LD cases and the mortality rate of 10% for community acquired cases, the mandatory annual *Legionella* monitoring has been extended not only to public building but also to those for commercial use and apartments [12,13]. In Flint, Michigan, the increased incidence of LD has drawn attention to contamination of domestic plumbing, although an epidemiological correlation with patient isolates has never been found [14,15].

In response to the need for prevention and control of *Legionella* contamination in buildings, the new Directive (EU) 2020/2184 of the European Parliament and of the Council on the quality of water intended for human consumption (December 2020) was recently published. For the first time *Legionella* was introduced as microbiological parameter to be monitored, adopting effective control and management measures proportionate to the risk in order to prevent further cases. To this end, the new directive establishes for *Legionella* the parametric value of 1000 CFU/L (the value that has been found in the apartments of the cases described here) to apply control measures considering also lower values when cases of infection or outbreaks occur.

The investigation in the building water system in which the cases lived highlighted some critical issues: the temperature of water sampled by the boiler and recirculating water system was 56 and 45 °C, respectively, and the boiler was located on the top floor of the building, without any isolation barrier. Consisting with the well-known ability of *Legionella* to grow at temperatures between 30 and 45 °C, the temperature of boiler was certainly a risk factor. Furthermore, as usually in Italy, a *Legionella* risk assessment plan is very rarely adopted for private homes or buildings.

Typing demonstrated that the building was colonized by *L. pneumophila* serogroup 1 ST23 that is known to be quite widespread in Italy where it caused both sporadic and epidemic cases [16,17,18,19]. The *Legionella* genome isolated from the patient was also sequenced as part of another study, and the comparison with other ST23 isolated from other cases, distant in time and space, showed that all were very similar (accession number: SAMN14088751; unpublished data). As shown by SBT database, ST23s are found primarily among community and travel associated cases where the risk of acquiring the disease is lower. In this study we did not analyze the whole genome of the ST23s isolated in the apartments but, since the patient’s ST23 was the same as that isolated during a previous outbreak in some patients’ homes [17], it could be hypothesized that also the environmental strains isolated here may have a very similar genome sequence.

Some outbreaks of LD that occurred in the past in Italy have also involved people who claimed to have never left their homes. Even in those cases, the source of infection was most likely the water system of their homes [16,17]. Moreover, in some homes the *Legionella* contamination level was quite high, and it is well-known that, although disinfecting treatments can maintain under control the risk of infection, the eradication is rather impossible. This study highlights the importance of prevention, control and monitoring of *Legionella* in private buildings/homes especially when elderly and immunocompromised patients are present. The imminent transposition of the new Drinking Water Directive into national legislation will be an important opportunity for a deeper and systematic investigation of premises’ plumbing systems for definitively addressing this risk which has been very often not adequately estimated.

## Figures and Tables

**Table 1 ijerph-18-06922-t001:** *Legionella pneumophila* concentration in the different sampling sites.

Site of Sampling	Identification	CFU/L
Shower apartment patient A	*L. pneumophila* sg1	4.7 × 10^3^
Bathtub head shower apartment patient A	*L. pneumophila* sg1	1.1 × 10^3^
Bathtub head shower apartment patient B	*L. pneumophila* sg1	6.4 × 10^3^
Shower apartment patient B	*L. pneumophila* sg1	1 × 10^3^
Boiler’s bottom of the building water system	*L. pneumophila* sg1	1.1 × 10^4^
Hot recirculating water of the building water system	*L. pneumophila* sg1	1.4 × 10^4^
Shower apartment patient C	ND	-
Municipal water network near the building	ND	-
Municipal water network	ND	-

ND: not detected; sg: serogroup.

## Data Availability

Data available in a publicly accessible repository.

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
