# Peer review of "A Legionnaires’ Disease Cluster in a Private Building in Italy"

_ijerph, 2021, doi:10.3390/ijerph18136922_

Round 1

Reviewer 1 Report

The brief report by Ricci et al describes two cases of Legionnaires’ disease in residents of the same apartment building in Italy. The authors use Mab and SBT analysis to show that an isolate recovered from one of the patients was of the same type (Philadelphia ST23) as isolates recovered from sample points in the apartment building’s water supply, thus confirming the likely source of the infections. The study is well designed and technically sound, however the English language used needs improving. A strength of the manuscript is the use of current gold standard typing methods (although soon to be superseded by whole-genome sequencing-based ones). The manuscript is also timely (as the authors mention) given inclusion of Legionella as a microbiological parameter to be monitored in a new EU Directive on water quality.        

Minor comments:

  1. The authors provide the CFU/L of L. pneumophila present in the nine water samples, but provide no information in the methods as to how these levels were determined.
  2. Introduction – Restriction fragment length polymorphism (RFLP), not RFPL.
  3. Results, please clarify if the environmental samples typed were ALL subgroup Philadelphia ST23 as this is unclear as written.
  4. Methods – the latex agglutination test (Oxoid) used – does this confirm serogroup 1 or just species? Please clarify.
  5. Discussion – the new EU directive establishes the threshold value of 1000 CFU/L to apply control measures. It would be worth noting that some of the CFU/L values determined in this study are equal to, or only just higher than this value. Does this suggest the directive should ideally have a lower threshold value to be effective at preventing disease? Say 100 CFU/L, for example?
  6. Discussion – was any action taken to decontaminate the building water supply as a consequence of this investigation?

Author Response

Reviewer 1

Minor comments:

1. The authors provide the CFU/L of L. pneumophila present in the nine water samples, but provide no information in the methods as to how these levels were determined.

Response 1: at the end of section 2.3. Environmental investigation was added the sentence: “All water samples were analyzed by culture according to the ISO 11731:2017, using filtration with washing procedure and considering the samples water with high back-ground”.

2. Introduction – Restriction fragment length polymorphism (RFLP), not RFPL.

Response 2: the acronym has been corrected

3. Results, please clarify if the environmental samples typed were ALL subgroup Philadelphia ST23 as this is unclear as written.

Response 3: yes, all environmental samples were subgroup Philadelphia ST23 and this was added.

4. Methods – the latex agglutination test (Oxoid) used – does this confirm serogroup 1 or just species? Please clarify.

Response 4: latex agglutination test allows to distinguish serogroup 1 by the other serogroups and/or species. In section 3. Results is specified that all Legionella isolates were serogroup 1.

5. Discussion – the new EU directive establishes the threshold value of 1000 CFU/L to apply control measures. It would be worth noting that some of the CFU/L values determined in this study are equal to, or only just higher than this value. Does this suggest the directive should ideally have a lower threshold value to be effective at preventing disease? Say 100 CFU/L, for example?

Responce 5: The EU directive establishes a general threshold value of 1000 CFU / L for the application of control measures. However, when infections or outbreaks occur, this level can be even lower. Furthermore, each Member State can decide, on the basis of the risk assessment, to independently consider the priority premises on which this risk should be assessed and even if lower values can be considered (for example in homes where there are immunocompromised patients or in hospitals).

6. Discussion – was any action taken to decontaminate the building water supply as a consequence of this investigation?

Response 6: Local Health Authorities recommended the disinfection of the system, the raising of the hot water temperature and provided indications for a correct management of the water system, but we don’t know if these actions were really adopted even though no more cases occurred. A sentence was added in discussion.

Reviewer 2 Report

The present study is really interesting as it draws the reader's attention to the reality of domestic Legionella contamination which could be fatal. This need cannot be emphasized enough and it is my hope that this paper can inspire the government and all stakeholders to attend to this crucial need.

Minor comments:

  1. The authors do need to go over the manuscript again fix English usage (e.g missing words) throughout
  2. Line 49: What disease are the authors referring to here? LD, I guess?
  3. In describing the cases (lines 78-82), do the authors know what floors the two patients lived in the building? It would really help the readers to paint a mental picture of how easy (or difficult) it might be for L. penumophila to spread throughout a building. 
  4. The major drawback in this study is that the authors have not done a whole-genome analysis of the sequence type isolated in the current study. Although this point has been noted by the authors, doing this extra analysis would have added even more value to this work. A lot of adaptation would have happened between 2005 and 2017 in the environmental strains that a whole-genome analysis could easily show.

Author Response

Reviewers 2

Minor comments:

  1. The authors do need to go over the manuscript again fix English usage (e.g missing words) throughout

Response 1: thank you for your suggestion, manuscript will be reviewed for English usage.

2. Line 49: What disease are the authors referring to here? LD, I guess?

Response 2: We think that the reviewer is referring to section 2.1 Cases, where we write “the woman did not have underlying diseases”. Here we mean that the woman was healthy, without comorbidity.

3. In describing the cases (lines 78-82), do the authors know what floors the two patients lived in the building? It would really help the readers to paint a mental picture of how easy (or difficult) it might be for L. penumophila to spread throughout a building. 

Response 3: yes, we know the floors the two patients lived and in section 2.1 Cases the information was added.

4. The major drawback in this study is that the authors have not done a whole-genome analysis of the sequence type isolated in the current study. Although this point has been noted by the authors, doing this extra analysis would have added even more value to this work. A lot of adaptation would have happened between 2005 and 2017 in the environmental strains that a whole-genome analysis could easily show.

Response 4: the reviewer is right and we know that from now on it will be increasingly necessary to have the whole genome sequence considering the changes that the strains can undergo and the much information this method provides.